# Utilization of adequately iodized salt and its associated factors in Tanzania rural areas: a case of Kilwa district, Lindi region, 2023

David Mahwera[1,2], Rose Msaki[3], Rogath Kishimba[2,4], Fatma Abdallah[3], Danford Mahwera[5], Vicent Assay[3], George Mrema [1,2,4]*, Geofrey Mchau[1,2,3], Germana Leyna[1,3] Theresia Ambrose[1]

1 Department of Epidemiology and Biostatistics, Muhimbili University of Health and Allied Sciences, Dar es Salaam, Tanzania, 2 Tanzania Field Epidemiology and Laboratory Training Program, Ministry of Health, Dar es Salaam, Tanzania, 3 Department of Community Health and Nutrition, Tanzania Food and Nutrition Centre, Ministry of Health, Dar es Salaam, Tanzania, 4 Epidemiology and Disease Control Section, Ministry of Health, Dodoma, Tanzania, 5 College of Engineering, University of Dar es Salaam, Dar es Salaam, Tanzania

* drgeorgemrema@gmail.com

## Abstract

This study assessed household utilization of adequately iodized salt and its associated factors in Kilwa district, Tanzania, where utilization remains low, especially in areas with local salt production. Using a cross-sectional design, 493 households were systematically sampled and interviewed, with 14 local salt producers purposively recruited for in-depth interviews; onsite iodine rapid tests and laboratory analyses determined salt iodine content. Results showed only 9.4% of households used adequately iodized salt. Factors significantly associated with utilization included family size of five or fewer members (AOR = 3.49; 95% CI: 1.62–7.54), good knowledge about iodized salt (AOR = 4.97; 95% CI: 2.04–12.11), storage of salt in dry areas (AOR = 4.44; 95% CI: 1.51–13.07), exposure of salt to sunlight (AOR = 0.29; 95% CI: 0.10–0.85), and salt staying less than two months (AOR = 2.34; 95% CI: 1.10–5.00). Key reasons for low availability of iodized salt included poor protection at production sites, supply of non-iodized salt, lack of training for local producers, community preference for non-iodized salt, and presence of multiple local salt producers. The findings indicate that the prevalence of iodized salt use is very low in Kilwa, with factors such as family size, knowledge, storage practices, sunlight exposure, and salt duration influencing utilization. The study recommends intensified government awareness campaigns to improve knowledge and practices related to iodized salt use, alongside interventions to enhance salt quality and availability at production sites.

## Introduction

Iodine is a micronutrient that is necessary at modest levels for both human and animal physiologic function. It is a crucial part of thyroid hormones, which are essential

**Data availability statement:** All relevant data are within the manuscript and its Supporting Information files.

**Funding:** This work is supported by the United States President's Emergency Plan for AIDS Relief (PEPFAR) through the Centre of Excellence in Health Monitoring and Evaluation, Mzumbe University under the U.S Centers for Disease Control and Prevention (CDC), Project Cooperative Agreement No: NU2GGH002292. Additionally, they provide support in the training of residents in Tanzania's Field Epidemiology and Laboratory Training Program.

**Competing interests:** The authors have declared that no competing interests exist.

for regulating metabolic rate, growth, and development of bodily structures as well as neuronal function and growth [1]. Iodine deficiency disorders are a major public health concern in Tanzania. Approximately 25% of the population was said to have an iodine deficiency condition, while 41% of people lived in regions that had an iodine shortage [2]. In order to meet iodine requirements the daily intake of iodine recommended by the National Research of the US National Academy of science was 40 μg/day for young infants (0–6 month), 50 μg/day for older infants (6–12 months), 60–100 μg/day for children (1–10 years) and 150 μg/day for adolescent and adult [3]. Nearly 2 billion individuals worldwide are still affected by iodine deficiency illnesses which put them at risk for cognitive impairment, stillbirth, and irreparable brain damage [1,4,5]. Individuals from similar populations in regions without severe iodine shortage may have intelligence quotient that are up to 13.5 points higher than those in places with severe iodine deficiency [6].

The main cause of thyroid pathology in Africa is dietary iodine insufficiency, which leads to a variety of iodine deficiency illnesses, including goiters, hypothyroidism, and mental retardation, the last of which presents the greatest risk to socioeconomic well-being. Iodine deficiency poses a threat to 350 million Africans at the very least. Goiters are estimated to affect 28.3% of Africans, and the continent accounts for around 25% of the world's iodine deficiency burden as measured by disability-adjusted living years (DALYs) according to estimates from the WHO [7]. In Tanzania especially in the Southern Highlands regions, the total goiter prevalence (TGP) was up to 90%, and 50% of schoolchildren had hypothyroidism [8]. The global strategy for the eradication of iodine deficiency disorder (IDD) was identified as universal salt iodization (USI), which aims to iodize all salt for human and animal consumption in order to ensure adequate iodine nutrition. This strategy has been proposed in numerous studies carried out in various locations [9]. Around the world, 70% of households consume adequate iodized salt, and more than 120 nations are running USI initiatives. Of these, 34 countries have already reached USI, another 28 are almost there, and additionally, 84 million children are shielded each year from the danger of IDD, with the number of nations where the issue is still present falling to 47 [10].

Salt production in Tanzania primarily involves two main methods: solar evaporation along the coastal belt and islands such as Pemba, where seawater is collected and allowed to evaporate naturally under the sun; and thermal or boiling techniques used inland in regions like Dodoma and Njombe, where brine is heated to extract salt. The process begins by channeling seawater or brine into salt beds or ponds, where it is left to evaporate over time, leading to the crystallization of salt. Once the salt crystals form, producers harvest and dry the salt before packaging. In Tanzania, approximately 422 salt production sites are dispersed mainly along the coastal regions, with small-scale producers constituting about 86% of the total. A typical local salt producer employs an average of 2–3 workers, mainly family members or seasonal laborers, with small-scale farms often operated by individual farmers or informal groups. Packaging methods vary, including simple polypropylene bags, laminated polythene, or paper bags, often imprinted with product details such as origin, date, and weight; however, many small producers lack standardized labeling. Distribution patterns

indicate that about 90% of the salt produced is sold locally within regions, primarily to wholesalers, retailers, and households, while some salt is exported to neighboring countries like Rwanda and Malawi. In terms of affordability, small-scale producers face challenges due to high production costs, limited economies of scale, and restricted access to affordable potassium iodate, which affects both the quality and cost of salt. Consequently, most producers sell their salt at prices influenced by these constraints, often making iodized salt less accessible or affordable for poorer households, highlighting the need for strengthening support systems and consolidation models to improve quality and market access (Unpublished TFNC report "*Mapping of the salt sector for a comprehensive plan towards consolidation model for sustainable universal salt iodation*.2019").

USI has been adopted in Tanzania as a means of eliminating IDD in the country. USI goal is to ensure that household coverage in Iodated salt utilization reaches 90%, however, Tanzania still have a coverage of 61.2% as per Tanzania National Nutrition Survey of 2018 [11]. Iodized salt that contain >15ppm and <40 ppm were considered adequately iodized for the prevention of iodine deficiency [6]. Low coverage of iodized salt may place the community a risk of irreversible brain damage, stillbirth, and cognitive impairment [1]. In Tanzania, it was reported that low iodine utilization is mainly in rural areas and might be caused by the low knowledge on the importance use of iodized salt utilization, inadequate availability of iodized salt in the community, price of iodized salt and cultural beliefs in the community (Tanzania Food and Nutrition Centre (TFNC), unpublished annual report of 2021). This study provides the current prevalence of iodized salt utilization at the household level in rural and explore the factors that influence the household level utilization of iodized salt in the community as well as the reasons for inadequate availability of iodized salt in the community among local salt producers in Kilwa, Tanzania.

Understanding the current prevalence of households utilizing adequately iodized salt is essential, as is identifying the factors associated with iodized salt utilization and the reasons behind its adequate availability in the community, especially among local salt producers in Kilwa. This information will enable the government, through TFNC, to design targeted interventions to improve household coverage of iodized salt and support local salt producers in enhancing the quality and availability of adequately iodized salt. While previous studies have explored salt iodization, there remains a notable research gap concerning the specific factors influencing iodized salt use among households and the role of local producers in Kilwa. Addressing this gap will help tailor more effective strategies to combat iodine deficiency in the region. To supplement existing literature, therefore, this current work assessed household utilization of adequately iodized salt and its associated factors in Kilwa district, which is among Tanzania rural areas, and this was performed by way of cross-sectional design across local salt producers.

## Study methodology

### Schematic overview of study program

The flow diagram below illustrates the sequential methodology steps that structured this study. It encompasses major stages including sampling design, questionnaire development, data collection, laboratory analysis, and data analysis, aligning with the objective to assess iodized salt utilization and production practices in Kilwa district. This approach uniquely combines both quantitative and qualitative methods to provide a comprehensive understanding of the problem, leveraging the strengths of each to compensate for potential limitations. The steps depicted in the diagram were validated through pre-testing of tools, training of data collectors, and quality control measures such as blinded testing and laboratory instrument calibration, ensuring robustness and reliability of the findings (fig 1).

### Questionnaire development process

The development of the research instruments involved a systematic process that integrated expert experience, literature review, and contextual relevance. Initially, a comprehensive review of existing questionnaires on salt iodization and related

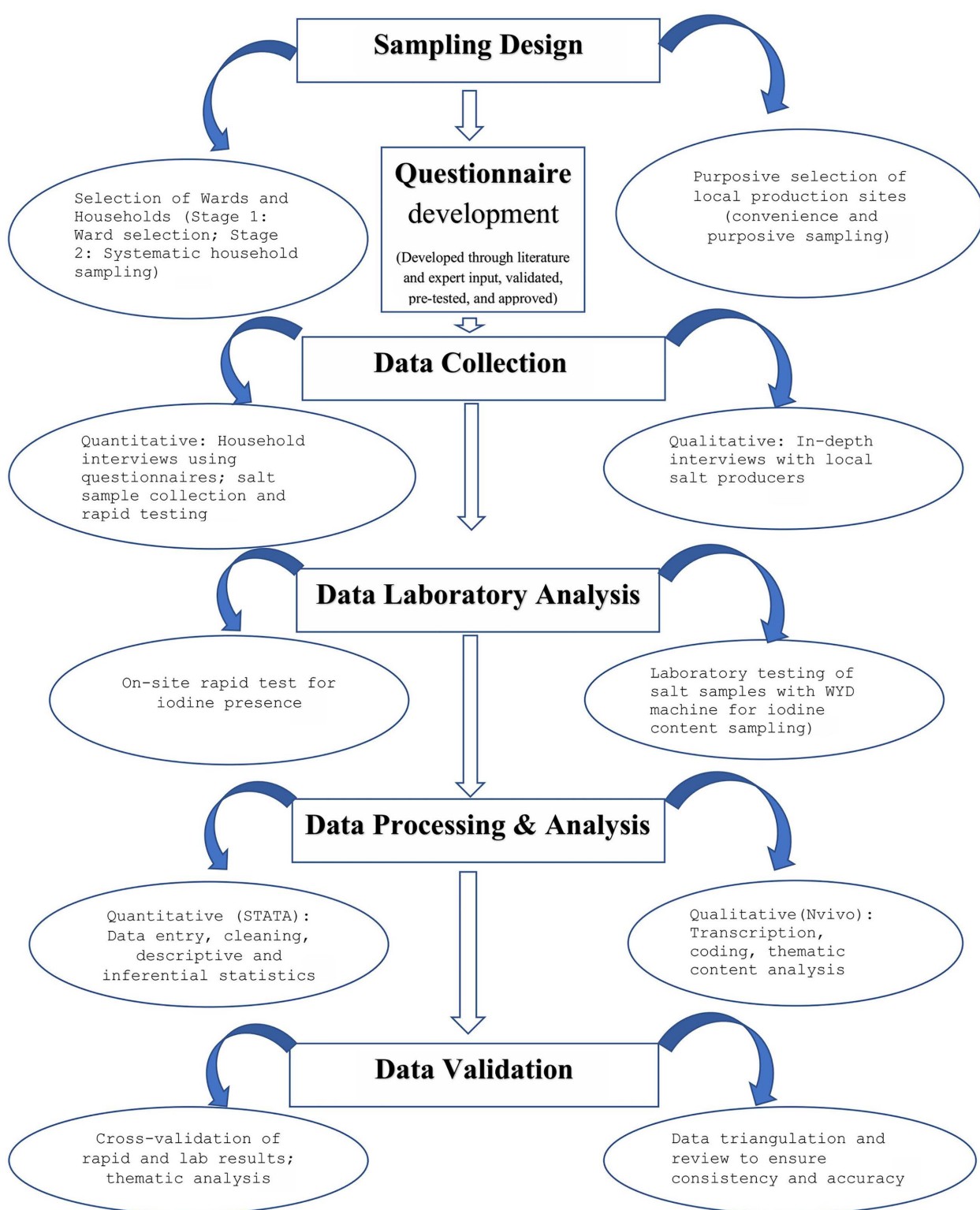

**Fig 1. The flow diagram showing schematic overview of study program.**

health behaviors was conducted, drawing insights from national and peer reviewed studies. To tailor the questionnaire to local context, we consulted with specialists in public health, nutrition, and salt production, whose expertise helped identify culturally appropriate and relevant items. The questionnaire contents were then drafted to include demographic information, salt storage practices, beliefs, knowledge, and consumption behaviors, ensuring all key variables were covered. A content validation process was carried out where subject matter experts reviewed the draft for clarity, relevance, and completeness; their feedback was incorporated, enhancing the instrument's validity. Subsequently, the finalized questionnaire was approved by the study supervisors and ethical review board.

The questionnaire was organized into 3 sections covering: (1) Socio-demographic characteristics, (2) Proportion of households using iodized salt, and (3) Factors associated with iodized salt utilization, including individual practices, beliefs about iodized salt, assessment, and purchase and accessibility factors. Covering 6 pages with a total of 30 questions, this study employed both structured and semi-structured questionnaires, both of which were pre-tested prior to the main data collection. During face-to-face interviews with household heads, both types of questionnaires were utilized; however, semi-structured questionnaires were primarily used to allow flexibility in probing responses and clarifying answers, thereby capturing more in-depth qualitative insights. The pre-testing involved a small sample of 12 household heads in a similar setting to assess clarity, relevance, and respondent understanding. This process helped identify ambiguous questions and logistical challenges, ensuring the validity and reliability of the main data collection instruments.

## Study design, area and population

A cross-sectional study was carried out in April 2023 in the Lindi region. Both quantitative and qualitative methods were utilized for data collection. The use of both quantitative and qualitative methods has a power to generate a clear picture of the problem under study than a single method can yield because weaknesses of one method can be compensated by another method [12]. Additionally, combination of both methods permits mutual validation of the findings [13]. Lindi region is located in southern part of Tanzania mainland having a population of 864,652 with an estimated growth rate of 0.9% and an area of 15,000 km$^2$ and, with 225,972 households (24). The region has one municipal council called Lindi municipal council and, five district councils namely Kilwa district council, Lindi district council, Liwale district council, Nachingwea district council and Ruangwa district council. The main source of income in Lindi region is agriculture [14]. Lindi was chosen because it has a lower proportion of households utilizing adequately iodized salt and the presence of local salt production sites Within Lindi region, Kilwa district was purposively chosen because it is the worst performer among all districts in Lindi region (Unpublished TFNC reports).

The study involved heads of households and local production sites in charges from Mandawa, Kiranjeranje, Miteja and Tingi wards.

## Sample size and sampling technique

We estimated the minimum sample size of 493 households using a formula for estimating a sample for proportions [15], considering previous proportion of households using adequately iodized salt at Lindi 37.6% [11]. A two-stage cluster sampling design was used for sample selection. We considered a design effect of 1.226, as reported in a national survey on iodine deficiency conducted in Tanzania among school children [11]. Additionally, we adjusted for a 10% non-response rate." In executing this sampling design, the first stage involved random selection of 4 wards (Mandawa, Kiranjeranje, Miteja, Tingi) from a list of available wards in Kilwa district and the second stage involved systematic selection of 493 households from 4 selected wards. Systematic sampling was ideal for our study due to the large population and availability of a comprehensive sampling frame. It ensures even distribution across the population, reducing clustering, and helps prevent selection bias through its structured, random approach. To ensure representativeness, the sampling frame was complete, the sample size was sufficient to capture population variability, and the interval ($k$) was appropriately calculated to ensure proportional representation throughout the population. Households were picked in the interval of 6 at Mandawa

ward, 5 at Kiranjeranje ward, 3 at Miteja ward and 3 at Tingi ward. The selection interval was obtained by dividing the total households in each selected ward by the desired sample size. The first house was picked randomly by spinning a pen then the following households were picked in north direction. Different selection intervals were used for different wards to account for variations in population size within each ward. This approach ensured proportional representation across wards, allowing for a more balanced and equitable sample distribution. By adjusting the selection interval based on the total number of households per ward, the sampling process-maintained consistency in coverage while avoiding over or under representation of any specific area

For the qualitative part of the study, a convenience sampling method was utilized to select 14 local production sites out of 101 sites present in Kilwa district. The choice was influenced by its practicality, ease of access, time efficiency and logistical constraints, such as limited funding and time However, this approach has potential limitations. It may introduce selection bias, as the selected sites might not be representative of all salt producers in the region. Consequently, the findings may have limited generalizability, reducing their applicability to the broader population. Despite these limitations, convenience sampling was considered appropriate for the exploratory nature of this study and for generating initial insights into the local production and availability of iodized salt.

Sites which were accessible and its hamlet leader was present in a day of interview was the one chosen for this study. It has been previously mentioned that at least 12 respondents are sufficient to reach data saturation in qualitative studies [16]. In our case qualitative data saturation for the in-depth interviews was determined after conducting nine out of the planned fourteen interviews, no new information was emerging. At this point, participant responses became repetitive and consistent with previously gathered data, indicating that thematic saturation had been reached. Although three additional interviews were conducted to confirm and strengthen the identified information, they did not yield any new information. This confirmed that the data collected after the ninth interview was sufficient to capture the range of perspectives relevant to the study objectives. Consequently, a purposive sampling approach was used to recruit 14 local production sites in charges. A purposive sampling aim to select participants who are more knowledgeable about the problem under investigation and was chosen based on years of experience in salt production [17].

## Data measurement and quality control

Data collection was conducted by trained research assistants under the supervision of the principal investigator from April 3–29, 2023, research assistants supported the quantitative part only, to avoid interview bias, the qualitative component was conducted by the Principal Investigator, who was skilled and experienced in conducting such interviews, ensuring consistency, neutrality, and professionalism throughout the data collection process. Prior to data collection the data collectors did not know the iodine content (whether it was high, low, or absent) of the salt samples they were collecting or testing. They were blinded, they treat all samples equally and record exactly what the test shows no assumptions.

Training was provided to all research assistants in the use of questionnaire and other data collection tools. Each data collector was provided with questionnaire in paper based, rapid test kits (RTKs) for testing salt samples on the site and plastic bags for salt sample collection and storage ready for laboratory analysis.

Semi Structured and pre-tested questionnaires were used to conduct face to face interviews with the heads of households to collect information on factors associated with adequately iodized salt utilization. However, an interview guide with open ended questions was used to conduct face to face in-depth interview with site in charges to collect information on the profile of the recruited local production sites at Kilwa district. In-depth interviews were audio recorded by a digital audio recorder for transcription, translation and coding.

Before starting the process of data collection, data collection tools (questionnaire and in-depth interview guide) were pre-tested among 12 number of head of households and 2 number of local production sites in-charges, in one of the wards namely Masoko ward within Kilwa districts which was not part of wards for this study. Masoko ward has similar characteristics.

Salt samples (a tea spoonful) collected from the households and local production sites were qualitatively tested for iodine using iodine rapid testing kit [18]. A salt samples which turned to blue or purple color after being exposed to a drop of test solution were considered to be iodized [19]. All salt samples which were positive for the rapid test were placed into a screwed capped plastic container and appropriately labeled ready for being transported to Kigoma Regional Referral Hospital (RRH) laboratory to be tested for iodine content. The laboratory analysis of these salt samples were performed by using WYD machine [20]. Before the sample being tested for iodine content, a WYD machine quality check was performed in Kigoma Regional Referral Hospital (RRH laboratory to ensure the accuracy and reliability of the instruments.

## Inclusion, exclusion criteria and variables

**Inclusion criteria.** The study recruited head of households and in charge of the local production sites lived in the study area for more than 12 months.

**Exclusion criteria.** Seriously ill participants and participants with mental problems were excluded to participate in the study.

**Dependent variable.** The dependent variable of this study was utilization of adequately iodized salt. This variable was used as a binary variable. Households that utilize salt with iodine content ≥15 ppm were termed to utilize adequately iodize salt and those with iodine content < 15 ppm were regarded to utilize inadequately iodized salt.

**Independent variables.** Independent variables included the following; Demographic and economic information such as age, sex, occupation, education level, and family income level. Household salt preference was assessed by identifying whether participants preferred rock salt or common/powdered salt. Beliefs regarding iodized salt were also explored, with households scoring "yes" if they reported cultural beliefs that prohibited the use of iodized. Practices related to iodized salt utilization were assessed through several factors, including the type of storage container used (container with lid, container without lid, or plastic bag), exposure to sunlight (yes or no), storage location (near fire, moist area, or dry area), place of purchase (open markets, salt farms, or local shops), and the duration the salt was kept at home (more than two months or two months or less). Additionally, accessibility was measured by the time taken to obtain salt (30 minutes or less versus more than 30 minutes). Economic factors, such as the perceived cost of iodized salt (cheap or expensive), and participants' knowledge about iodized salt were also assessed.

In this study Knowledge was assessed by calculating the mean score from four sets of questions designed to evaluate participants' knowledge by assigning 1 point to each correct answer, and 0 to an incorrect answer. First, the total number of correct answers was calculated for each participant. Then, the mean score was obtained by summing the correct answers of all participants and dividing by the total number of participants. Participants who scored greater than or equal to the mean were classified as having good knowledge, while those who scored below the mean were classified as having poor knowledge. Furthermore, income level was classified based on data from the Household Budget Survey (HBS) conducted by the National Bureau of Statistics (NBS) to assess income levels and poverty. This survey collects detailed information on household income and expenditure, which helps in understanding the distribution of income across different regions and demographics [21].

## Data processing and analysis

Quantitative data was collected by using questionnaires in paper form, was then entered in excel for cleaning and then transported to STATA version 15 for analysis. In this study both descriptive and inferential statistics were used to analyze quantitative data. All data are summarized in tables and figures for clear understanding. Bivariable and multivariable logistic regression were used to find the association between the dependent and independent variables. During bivariable analysis, variables with $p \leq 0.2$ were included in multivariable analysis. To compensate for the potential confounding influence of independent factors, a multivariable analysis was performed. In multivariable analysis, variables with $p \leq 0.05$ was considered significantly associated with utilization adequately iodized salt [22].

 

Qualitative data was audio recorded. Content analysis approach was used to analyze qualitative data using NVivo software [23] NVivo was chosen because it is user friendly, there was access to experts who could provide support during the analysis process, and we preferred the use of commercial software. During the content analysis approach, the researcher first transcribed all recorded information into word document. During this process the researcher ensured that all transcribed data are clean by eliminating all irrelevant information. Thereafter, transcribed data was translated into English language and imported into NVivo software, carefully coded and finally organized into appropriate themes [24].

## Definition of terms

**Adequately iodized salt.** Salt that contains ≥15 parts of iodine per million parts of salt, assessed by taking salt samples that become positive to RTK on site and sent in laboratory for quantitative analysis [25].

**Partially iodized salt.** Salt that contains less than 15 parts of iodine per million parts of salt assessed by taking salt samples that become positive to RTK on site and sent in laboratory for quantitative analysis.

**Non-iodized salt.** Salt that has no iodine which may cause a lot of public health problems such as congenital malformations, stillbirth, brain damage, etc. that was assessed by testing salt sample on site by rapid test kit (RTK).

**Utilization of adequately iodized salt.** This is the act of using/uptake or eating salt that contains ≥15 parts of iodine per million parts of salt. This study regarded participants with adequately iodized salt at home as participants who utilize adequately iodized salt.

**Household.** As per this study household is a group of individuals who live together and share the same cooking and living space [11].

**WYD Machine.** An instrument for quick measurement of Iodine concentration in Iodized salt [26]

## Ethical clearance

Ethical approval was obtained from the ethical review board at Muhimbili University of Health and Allied Sciences with approval number MUHAS-REC-11-2022-1462. All Participants for both qualitative and quantitative parts were given explanation on the objectives of the study then written and verbal informed consent was given to the participants that highlighted the right to withdraw from the study any time without consequences; only consented participants were involved in the study. Name and other personal identifiers were excluded in the questionnaire and only identification numbers were used instead to ensure confidentiality. During in-depth interviews, given the sensitivity of some responses, no participant was asked to provide their name, and each was assigned a unique ID number. All interviews were conducted individually at participants' respective farms to maintain privacy and avoid group influence. To ensure potential risks to participants were minimized, especially for local salt producers, care was taken to clarify that the study was not regulatory in nature and that the information collected would not be used for enforcement or inspection purposes. This helped reduce any fear of reprisal or negative consequences. Additionally, data collection on local salt producers was done by Principal investigator that had knowledge to uphold ethical principles, including respect, privacy, and cultural sensitivity throughout the study process. Additionally, permission to enter into selected wards was obtained from Ward Executive Officers (WEOs).

## Results

### Socio-demographic and economic characteristics

Out of the 493 visited households, 435 heads of households were available to participate in the study, resulting in a response rate of 88.2%. The characteristics of these respondents are provided below. Two hundred and seventy-three (62.8%) of the respondents were female. Regarding age, majority 252(57.9%) of the respondents were in the age group of

26–35 years with a mean age of 21 years (Standard deviation = 11). Majority 223 (51.3%) of the respondents had primary level of education, and 344 (79.1%) had low level of income (less than Tshs. 100, 000 per month). Peasants contributing to 239(54.9%) and households with more than five family members had 239(54.9%) of the total respondents. Respondents were drawn from 13 villages across four wards within Kilwa District, ensuring broad geographic representation of the study area. Table 1 summarizes the findings.

**Proportion of households that use adequately iodized salt**

Out of 435 collected salt samples from surveyed households, only 41 (9.4%) households were found utilizing salt with adequate iodine content (≥ 15 ppm) while 394 (90.6%) households had inadequately iodized salt (< 15 ppm) (Fig 2).

The above bar chart illustrates the distribution of iodine concentration levels in salt samples collected during the study. Two categories are represented: salt with adequate iodine levels (≥ 15 parts per million [ppm]) and salt with inadequate iodine levels (< 15 ppm). The chart shows that only 9.4% of the salt samples met the recommended iodine concentration for human consumption, while a significantly larger proportion, 90.6%, had iodine levels below the recommended threshold. This indicates a major public health concern regarding the availability and use of adequately iodized salt in the study area.

Stratification by ward revealed substantial variation in the use of adequately iodized salt. Miteja ward 17 (41.5%) had the highest proportion of households with adequately iodized, whereas Kiranjeranje ward 3 (7.3%) recorded the lowest. A similar pattern was observed at village level, where Njia Nne and Sinza villages had the highest adequacy levels (9 (21.9%) each), while several villages including Mbwemkuru, Mtandi, and Kiswele reported no household using adequately iodized salt. Fig 3 and 4 provide more details.

Further analysis of household demographic characteristics with respect to age and sex, adults aged 26–35 constituted the largest proportion of respondent using adequately iodized salt. Utilization was similar between males and females across most age categories (fig 5). Furthermore, respondent with primary education level 21 (51.2%) accounted for the majority of adequate users (fig 6).

**Knowledge on iodized salt**

Table 2 shows that about 50.6% of the participants had good knowledge about iodized salt. Specifically, majority (82.5%) heard about adequately iodized salt. Of all heard about adequately iodized salt, majority 43.7% got information from health workers. Regarding salt identification, majority (81.2%) had no knowledge for identification of iodized salt and 68.3% knew the benefits of using adequately iodized salt. Proportion of respondents with knowledge on the effect on non-use of adequately iodized salt (44.4%) was nearly equal to those who had no such knowledge (55.6%).

**Factors influencing utilization of adequately iodized salt**

Table 3 shows that of all variables analyzed during binary logistic regression only gender, family size, cultural belief, knowledge about iodized salt, storage place, expose to sunlight, time to stay with salt and perceived price of iodized salt are associated with utilization of adequately iodized salt. However, during multivariable logistic regression, family size, knowledge about iodized salt, storage place, expose to sunlight and time to stay with salt significantly influence utilization of adequately iodized salt.

The odds of utilization of adequately iodized salt are 3.49 (AOR, 3.49; 95%CI: 1.62, 7.54) times higher among households with family size with equal or less than five members compared with households with more than five family members. Moreover, the odds of utilization of adequately iodized salt among households with good knowledge about iodized salt are 4.97 (AOR. 4.970; 95%CI: 2.04, 12.11) times higher compared with households with poor knowledge. Additionally, households which store their salt in dry area are 4.44 (AOR, 4.44; 95% CI: 1.51, 13.07) times more likely to utilize adequately iodized salt than households which store their salt near fire. Again, the odds of utilization of adequately iodized salt are decreased by 70% (AOR, 0.30; 95%CI: 0.10, 0.85) among households which expose their salt at sunlight compared to

**Table 1. Socio-demographic and economic characteristics of the respondents in Kilwa District, Lindi Region (n = 435).**

| Variable | Category | Frequency (%) |
|---|---|---|
| Gender | Female | 273 (62.8) |
| | Male | 162 (37.2) |
| Age Group (Years) | Below 18 yrs. | 16 (3.7) |
| | 18-25 | 28 (6.4) |
| | 26-35 | 252 (57.9) |
| | 36-45 | 74 (17.0) |
| | Above 45 yrs. | 65 (14.9) |
| Level of Education | Illiterate | 115 (26.4) |
| | Primary Education | 223 (51.3) |
| | Secondary Education | 79 (18.2) |
| | Higher Education | 18 (4.1) |
| Source of income | Government Employees | 13 (3.0) |
| | Merchants | 52 (12.0) |
| | Self-employed | 131 (30.1) |
| | Peasants | 239 (54.9) |
| Income level | Low | 344 (79.1) |
| | Average | 86 (19.8) |
| | High | 5 (1.1) |
| Family size | >5 members | 239 (54.9) |
| | ≤5 members | 196 (45.1) |
| Village | Hotel tatu | 54 (12.4) |
| | Kibaoni | 11 (2.5) |
| | Kiranjeranje | 69 (15.9) |
| | Kiwawa | 15 (3.5) |
| | Mandawa | 76 (17.5) |
| | Matandu | 28 (6.4) |
| | Mbwemkuru | 21 (4.8) |
| | Mtandango | 19 (4.4) |
| | Mtandi | 16 (3.7) |
| | Njia nne | 46 (10.6) |
| | Sinza | 35 (8.1) |
| | Kiswele | 19 (4.4) |
| | Mitumba | 26 (6.0) |
| Ward | Kiranjeranje | 125 (28.8) |
| | Mandawa | 171 (39.3) |
| | Tingi | 65 (14.9) |
| | Miteja | 74 (17.0) |

non-expose to sunlight. Lastly, households which stay with salt for less than two months are 2.34 (AOR, 2.34; 95% CI: 1.10, 5.00) times more likely to utilize adequately iodized salt than households which stay with salt for more than two months.

### Reasons for inadequate availability of adequately iodized salt

**Poor protection of salt production sites.** Low availability of adequately iodized salt in the study area was also caused by poor protection of salt production sites as one of the respondents mentioned that:

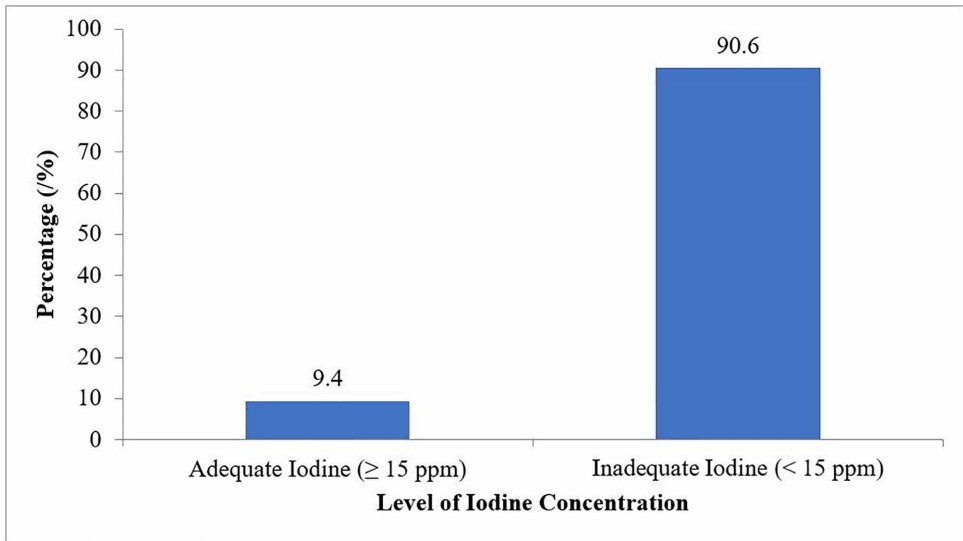

**Fig 2. Level of iodine content in the collected salt samples in households at Kilwa district, 2023.**

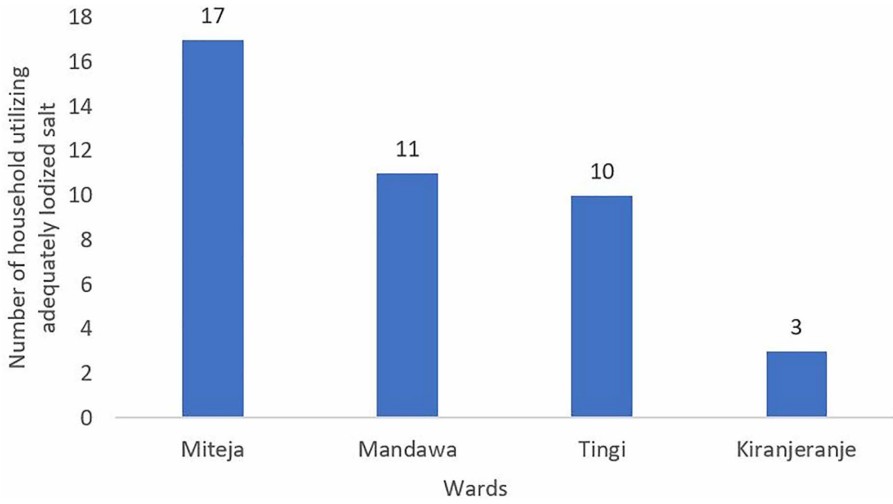

**Fig 3. Adequacy of iodized salt use among households by wards, Kilwa District, 2023.**

*…after collecting it in the fields because there are no guards, people go there and steal as much as they see fit* (Respondent 2).

Similar to this opinion, another respondent argued that:

*…the salt without iodine is often stolen and people take it to the street to sell* (Respondent 3).

**Supply of non-iodized salt to workers.** Findings from interview also revealed that a tendency of in charges of local production sites to provide non-iodized salt to their workers free of charge contribute to poor utilization of adequately

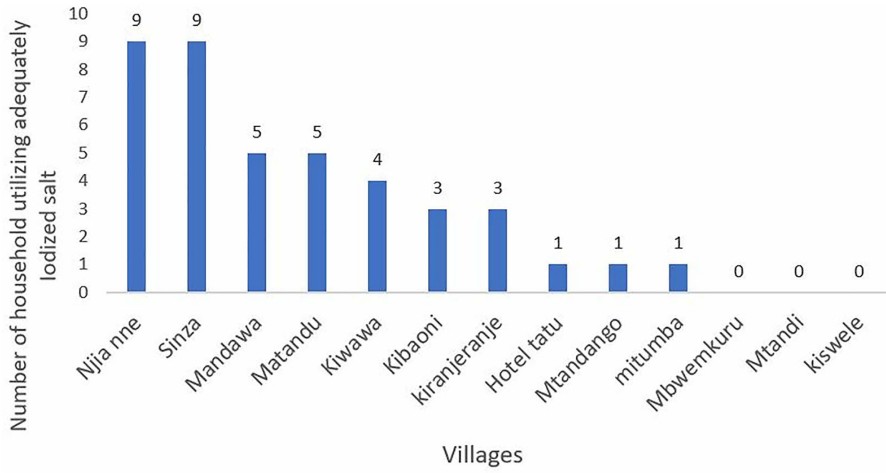

**Fig 4. Adequacy of iodized salt use among households by village, Kilwa District, 2023.**

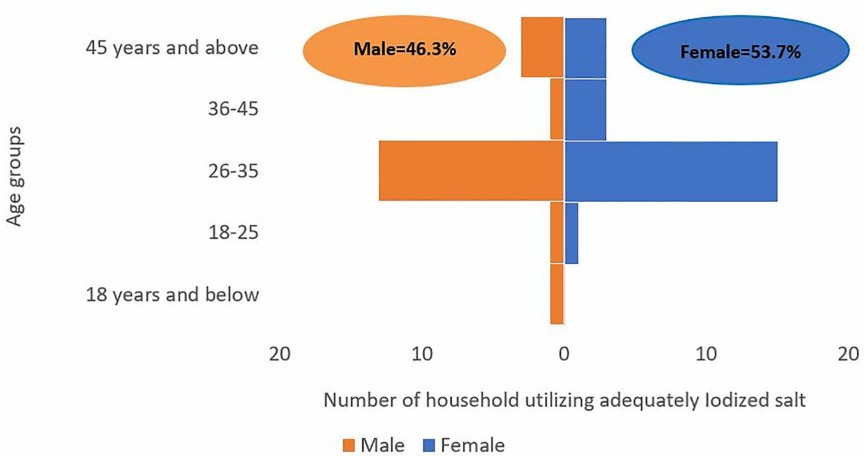

**Fig 5. Distribution of adequately iodized salt use by age group and sex among surveyed households, Kilwa District, 2023.**

iodized salt in Kilwa district. This was revealed by the majority of the participants. One of the respondents had the following to say during interview session:

*…No, we don't give salt as wages, but when we finish harvesting, they are given salt for use in their homes* (Respondent 4).

Similar view was given by another respondent who asserted:

*…Yes, sometimes when there is not enough money, we usually give them about two sacks of salt to sell and earn money, we don't put iodine in salt until we are transporting it, so they get salt without iodine* (Respondent 5).

**Unavailability of training to local salt producers.** Absence of formal training to local salt producer was another reason for poor availability of adequately iodized salt in the study area as stated by the majority of the participants. During interview session one the respondent had the following to say:

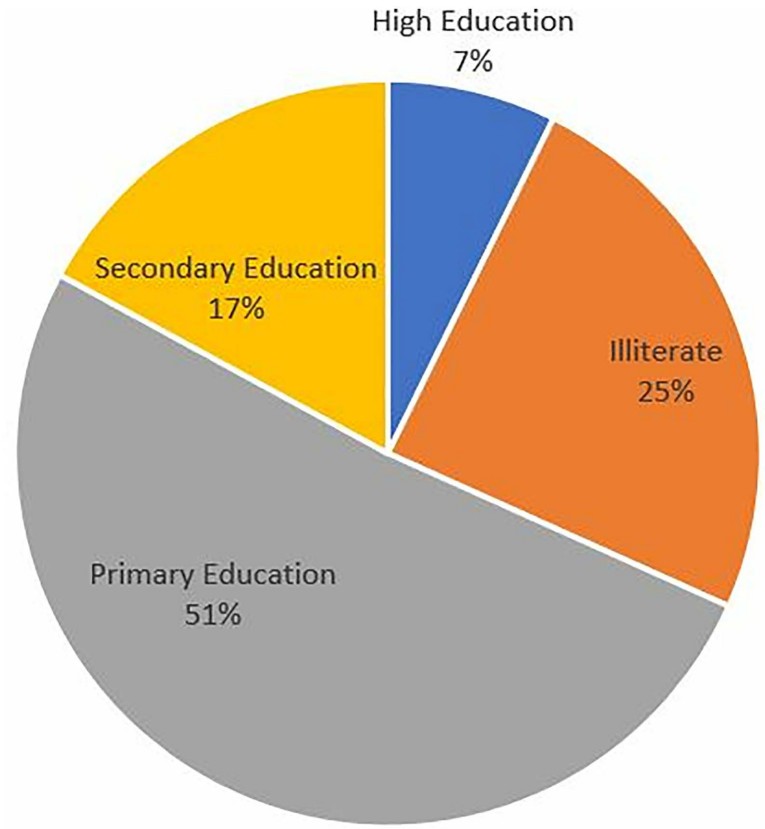

**Fig 6. Distribution of adequately iodized salt use by education level of household respondents, Kilwa District, 2023.**

**Table 2. Knowledge on iodized salt among households in Kilwa district, 2023.**

| Variables | Category | Frequency (%) |
|---|---|---|
| Overall level of knowledge about iodized salt | Poor | 215 (49.4) |
| | Good | 220 (50.6) |
| Heard about adequately iodized salt | No | 76 (17.5) |
| | Yes | 359 (82.5) |
| Source of Information | Mass media | 21 (5.8) |
| | Health workers | 157 (43.7) |
| | Friends and relatives | 92 (25.6) |
| | Salt sellers | 89 (24.8) |
| Ability to know if salt is iodized | No | 354 (81.2) |
| | Yes | 81 (18.6) |
| Ability to identify benefits of using adequately iodized salt | No | 138 (31.7) |
| | Yes | 297 (68.3) |
| Ability to identify effect on non-use of adequately iodized salt | No | 242 (55.6) |
| | Yes | 193 (44.4) |

**Table 3. Factors associated with adequate iodized salt utilization among households in Kilwa district, 2023 (n = 435).**

| Variables | Total | Yes | Crude odds ratio (95% CI) | Adjusted odds ratio (95% CI) |
|---|---|---|---|---|
| **Gender** | | | | |
| Female | 273 | 22 (8.1%) | 1 | |
| Male | 162 | 19 (11.7%) | 1.52 (0.79, 2.90) | 1.81 (0.84, 3.94) |
| **Age** | | | | |
| Below 18 | 16 | 1 (6.3%) | 1 | |
| 18-25 | 28 | 2 (7.1%) | 1.15 (0.10, 13.82) | |
| 26-35 | 252 | 28 (11.1%) | 1.88 (0.24, 14.74) | |
| 36-45 | 74 | 4 (5.4%) | 0.86 (0.09. 8.22) | |
| Above 45 | 65 | 6 (9.2%) | 1.53 (0.17, 13.65) | |
| **Level of Education** | | | | |
| Illiterate | 115 | 10 (8.7%) | 1 | |
| Primary education | 223 | 21 (9.4%) | 1.09 (0.50, 2.40) | |
| Secondary education | 79 | 7 (8.9%) | 1.02 (0.37. 2.81) | |
| Higher education | 18 | 3 (16.7%) | 2.10(0.52, 8.51) | |
| **Family size** | | | | |
| >5 members | 239 | 12 (5.0%) | 1 | |
| ≤5 members | 196 | 29 (14.8%) | 3.29 (1.63, 6.63) | 3.49(1.62, 7.54) ** |
| **Income source** | | | | |
| Government | 40 | 6 (15.4%) | 1 | |
| Merchants | 209 | 19 (13.5%) | 0.86 (0.16, 4.70) | |
| Self-employed | 130 | 10 (8.4%) | 0.50 (0.10, 2.57) | |
| Peasants | 56 | 6 (8.8%) | 0.53 (0.11, 2.55) | |
| **Income level** | | | | |
| Low | 344 | 30 (8.7%) | 1 | |
| Average | 86 | 10 (11.6%) | 1.38 (0.65, 2.94) | |
| High | 5 | 1 (20.0%) | 2.62 (0.28, 24.17) | |
| **Salt preference** | | | | |
| Common salt | 278 | 28 (10.1%) | 1 | |
| Rock salt | 157 | 13 (8.3%) | 0.81 (0.41, 1.61) | |
| **Cultural beliefs** | | | | |
| Yes | 216 | 14 (6.5%) | 1 | |
| No | 219 | 27 (12.3%) | 2.03(1.03, 3.99) | 1.89 (0.87, 4.10) |
| Variables | Total | Yes | Crude Odds Ratio (95% CI) | Adjusted Odds Ratio (95% CI) |
| **Knowledge about iodized salt** | | | | |
| Poor | 215 | 7 (3.3%) | 1 | |
| Good | 220 | 34 (5.5%) | 5.43 (2.35, 12.55) | 4.97 (2.04, 12.11) * |
| **Storage place** | | | | |
| Near fire | 111 | 5 (4.5%) | 1 | |
| Moist area | 189 | 10 (5.3%) | 1.18 (0.39, 3.56) | 1.19 (0.37, 3.82) |
| Dry area | 135 | 26 (19.3%) | 5.06 (1.87, 13.66) | 4.44 (1.51, 13.07) ** |
| **Type of storage container** | | | | |
| Container with lid | 50 | 7 (14.0%) | 1 | |
| Plastic bag | 303 | 27(8.9%) | 0.60 (0.45, 1.47) | |
| Container without lid | 82 | 7 (8.5%) | 0.57 (0.19, 1.74) | |
| **Expose to sunlight** | | | | |
| No | 318 | 36 (11.3%) | 1 | |

*(Continued)*

**Table 3.** (Continued)

| Variables | Total | Yes | Crude odds ratio (95% CI) | Adjusted odds ratio (95% CI) |
|---|---|---|---|---|
| Yes | 117 | 5 (4.3%) | 0.35 (0.13, 0.91) | 0.30 (0.10, 0.85) ** |
| **Duration you stay with salt** | | | | |
| > 2 months | 262 | 14 (5.3%) | 1 | |
| ≤ 2 months | 173 | 27 (15.6%) | 3.28 (1.66, 6.45) | 2.34 (1.10, 5.00) * |
| **Salt buying place** | | | | |
| Open Market | 181 | 17 (9.4%) | 1 | |
| Salt farms | 88 | 9 (10.2%) | 1.10 (0.47, 2.58) | |
| Local shops | 166 | 15 (9.0%) | 0.96 (0.46, 1.99) | |
| **Distance to get salt** | | | | |
| ≤30 minutes | 368 | 34 (9.2%) | 1 | |
| >30 minutes | 67 | 7 (10.4%) | 1.15 (0.49, 2.71) | |
| **Perceived cost of iodized salt** | | | | |
| Cheap | 49 | 8 (16.3%) | 1 | |
| Expensive | 386 | 33 (8.5%) | 0.48 (0.21, 1.11) | 0.82 (0.31, 2.47) |

*P-value<0.05, **p-value<0.01, ***p-value<0.001 Significant on multivariable logistic regression

*For about 3 to 4 years there has been no training in salt production or handling*" (Respondent 1).

Similarly, another respondent highlighted that:

*…There is no special training, but we teach each other in the field. We have never been given formal education from the district level or other lower levels*" (Respondent 7).

**Presence of other local salt producers in the community.** It was found that in the community there some people who are producing salt at their home in a small amount which was not enough to be transported outside the village and hence they just produce for selling in the local market and using at home

"*… some people cook salt in their homes for their own use and sometimes they sell it to people in the village*" (Respondent 2).

**Community preference on non-iodized salt.** The study also reviled that the majority in the community including the local salt producers want/prefers salt that is straight coming from the farm which is not added with anything including iodine(non-iodized)

"*However, people don't want industrial salt, they demand it contains chemicals, that's why they fail to reproduce, people believe that salt like salt from the sea is good and it gives you strength*" (Respondent 2).

Similarly, another respondent highlighted that:

"*... although the community and homes do not want salt with iodine*" (Respondent 1).

## Discussion

Findings of this study revealed that 9.4% of households in Kilwa district utilize adequately iodized salt (≥ 15 ppm) which is lower than 90% recommended by WHO. Based on these findings, one may argue that despite of existence of universal

salt iodization programme in the country a large number of households in Kilwa district are at high risk of developing iodine deficiency illnesses. Furthermore, although there is a legal framework under the Tanzania Food, Drugs and Cosmetics Act (CAP 219) specifically the Iodated Salt Regulations, published on 23 April 2010 which governs the production, importation, and distribution of iodated salt, challenges still persist. These regulations detail several key requirements, including restrictions on the importation of iodated salt, standards for edible salt, conditions for packaging and labeling, compliance with packaging and storage guidelines, inspection and analysis procedures by authorized officers, and penalties for non-compliance [27]. Low availability of adequately iodized salt in the study area might be attributed to households' poor salt storage practice and/or poor technology employed by local salt producers. The proportion of households with adequately iodized salt found in this study is lower compared to studies conducted in Tach Armachio and Ahferom districts in Northern Ethiopia which were 61.1% and 17.5%, respectively [28,29]. Additionally, this value is also lower than 75% in a study conducted in Prakasam district, India [30]. Moreover, the observed prevalence of utilization of adequately iodized salt in the study area is still lower than recently reported values in two Himalayan districts; 89% in Una district and 91% in Hamirpur district [31]. Due to low utilization of adequately iodized salt contributes to iodine deficiency disorders (IDD), including goiter problems [2]. To address this problem, it is advised to Increase awareness and education on the importance of adequately iodized salt utilization, ensure the consistent availability and accessibility of adequately iodized salt in the community by Working with local salt producers, suppliers, and retailers to ensure a reliable supply chain that can distribute iodized salt to households and local markets. By implementing these solutions, it is possible to increase the utilization of adequately iodized salt, reduce the prevalence of iodine deficiency disorders, and improve the overall health and well-being of the community. Tanzania's past interventions offer a successful model. Following a 1990 feasibility study, the government supported salt producers by distributing 72 iodation machines, providing training on their use, and establishing quality control systems. A national salt monitoring system was launched, supported by iodine testing laboratories and legal enforcement by trained health inspectors. These efforts boosted iodated salt supply from nearly zero to 18.6% of national demand by 1992. Salt iodation became legally mandated in 1995, significantly increasing access and demonstrating that collaboration with salt producers can greatly improve the availability and use of iodized salt in the country [3]

Findings of this study indicated that, household with equal or less than five members was positively associated with utilization of adequately iodized salt compared to households with more than five family members (Table 3). This is possibly due to the high price of iodized salt at the study area, hence, as a family becomes large, it opts to purchase cheaper salt which is mostly non-iodized. This finding aligns with a study done in Dire Dawa City in Ethiopia which reported that utilization adequately iodized salt was reduced to 61% for households with more than five members [32]. Education on the importance of using adequately iodized salt is very key to the change toward utilization and attaining USI goal, supporting local producers will Encourage local production of iodized salt, either through support for small-scale salt producers or by establishing community-based iodization programs. This can help increase the availability of adequately iodized salt at a more affordable price, benefiting households with larger family sizes.

Moreover, the odds of utilization of adequately iodized salt among households with good knowledge about iodized salt are 4.97 times higher compared with households with poor knowledge (Table 3). This might be due to the fact that households with good knowledge regarding the effect of non-use of adequately iodized salt help them to make good decision when purchasing salt. The finding of this study is consistence with those reported in Gidami district, Ethiopia which indicated that the utilization of adequately iodized salt was higher among households with good knowledge on iodized salt [33].

This study revealed that, the odds for utilization of adequately iodized salt among households storing their salt in dry areas are higher compared to those storing their salt near fire (Table 3). This finding concurs with a study done in Dera district in Ethiopia whereby the ability of salt stored in dry areas to retain iodine content was higher than salt stored near fire areas [34]. Findings of this study also indicated that, utilization of adequately iodized salt were 70% lesser among households exposed their salt to sunlight compared to those who did not (Table 3). Again, this finding aligns with a study done in Tach Armachio district in Northern Ethiopia which reported that utilization of salt with adequate iodine content was

reduced to 61% for households exposed their salt to sunlight [28]. It has been mentioned that salt tends to lose iodine when stored in areas such as heat, moisture and light during storage [35].

Lastly, time to stay with salt was also found significantly associated with utilization of adequately iodized salt (Table 3). It was revealed that, as household stay with salt for less than two months, the chance of utilizing adequately iodized salt increase. Findings of this study is consistence with a study carried out in Dera district, Ethiopia in which the availability of adequate iodine content was more likely to be found in salt samples stored for less than two months compared to salt samples stayed for more than two months [34].

This study revealed that poor protection of salt production sites was among the reason that hindered the availability of adequately iodized salt in the community. This study identified and observe the presence of salt sites without being protected as a results salt production sites are vulnerable to unauthorized access and theft. Individuals or groups may exploit the lack of security to steal salt which is not iodized and take home for their use or to sell at their local market leading to a shortage in the supply of adequately iodized salt to the community. This was contrary to the study conducted in Madagascar that require the production sites to be registered before commencing the production, the study revealed that before a place of production is registered or given license to function, it must be examined using a certification checklist to guarantee that the "must have items" essential for the manufacture of adequately iodized salt are present. The problem of production site protection was addressed in the check list by assuring the existence of "fence or wall and clean from rubbish and debris, animals, and unauthorized person passage are assessed. The study in Madagascar revealed that their salt production site was well protected [36]. Poor protection of salt production sites in Kilwa district might be attributed to lack of awareness among local salt producers as local producers may not fully understand the importance of protecting salt production sites and the implications of inadequate security measures. They may not be aware of the potential risks, such as theft, contamination, or adulteration, and how these can affect the availability of adequately iodized salt. Also, it might be due to limited resources as local salt producers, especially small-scale ones, may have limited financial resources to invest in security measures and lastly it might be due to the weak enforcement and regulation as there might be a lack of strict enforcement and regulation by local authorities or government agencies regarding the protection of salt production sites.

Additionally, it was found that workers were given salt to use in their homes, but in some sites, the amount provided was intended for them to sell in the market. The results from this study contradict with Nepal iodized salt act which prohibit the importation or distribute iodine free salt to the community [37]. This practice can be attributed to the owners or site supervisors attempting to compensate the workers by providing salt as a form of wages that they can sell to earn money. This situation typically arises when there is a shortage of customers to purchase the harvested salt, resulting in a lack of funds to pay the workers for their services. To address this issue, the owners or site supervisors opt to give bags of salt to the workers to sell on their behalf. However, since iodine is typically added during transportation or when customers are present, the salt provided for sale is often non-iodized. This practice not only compromises the availability of adequately iodized salt in the market but also poses a risk to the workers who rely on the income generated from selling the salt. It perpetuates the cycle of non-iodized salt consumption and contributes to the overall low utilization of adequately iodized salt in the community.

Moreover, this study found that there is unavailability of training to local salt producers in Kilwa district. These findings were in line with study another conducted in Tanzania which also indicated that unavailability of training caused production of inequality salt a situation which was improved by giving training to local salt producers [38]. The training helps the local salt producers to know the importance of adding iodine to the salt, proper storage as well as the right concentration to add. The lack of training in Kilwa might be attributed by the absence of coordination and collaboration among different stakeholders involved in salt production and iodization.

It is anticipated that all salt producers in the community were supposed to be recognized by the authority (Kilwa District Health Management Team) but this study revealed the presence of other local salt producers in the community. These were the people who were producing salt at their home place and were not recognized by the authority. It was found that

they were producing the amount that was insufficient to be transported outside the district but enough to be taken to their local markets. These producers don't add Iodine to the salt they produce as a result they tend to increase the problem of inadequate availability of adequately iodized salt in the community. The presence of these local salt producers might be caused by the difficulty in life as a result they try to produce salt for use in their homes and generating income for their families by selling the salt they produce. Despite of hardships of life also lack of knowledge about iodized salt led to these people to produce and sell salt without adding iodine.

Lastly, the study revealed that the community prefers the use of non-iodized salt over iodized salt. This was another reason given by local salt producers to why there is non-iodized salt in the community, they say people in the community (Kilwa district) including themselves prefer to use salt that is directly coming from the farm they believe it is not added with anything and it is pure. This is contrary to the study conducted in Dessie and Combolcha Town, South Wollo, Ethiopia and the study done in Prakesam district in India which revealed that the majority (80.4%) and (83.6%) of participants respectively prefer iodized salt over non iodized salt [18,30]. The preference of non-iodized salt over iodized salt might be due to the presence of cultural believes in the community that they believe iodized salt is not good for their health and might cause sterility to both men and women. Due to this, local producer has no other options other than supplying non-iodized salt to meet the demand. Other reasons might be due to poor knowledge about iodized salt among both local salt producers and community as a result they don't know the benefits and consequences of supplying and using non-iodized salt in their community. Given the cultural preference for non-iodized salt observed in the study area, it is important to implement culturally sensitive education strategies that directly address local beliefs and misconceptions. These strategies should involve community leaders, traditional healers, and respected elders, who can serve as trusted messengers. Educational campaigns should use local languages and culturally appropriate messaging, highlighting the health benefits of iodized salt in the context of local practices and values. Interactive approaches such as community dialogues, storytelling, radio programs, and drama-based outreach can help demystify misconceptions and encourage behavior change. Involving women's groups and school-based programs may also be effective, as women and children are often key decision makers and vulnerable populations. Tailoring the content to align with local customs while reinforcing scientific facts will enhance acceptance and promote sustainable use of iodized salt.

## Challenges and limitations of current study

This study faced several challenges and limitations. One major constraint was the limited availability of prior research on factors influencing iodized salt utilization. Most of the existing literature originated from a single country, i.e., Ethiopia which restricted the diversity of evidence available to inform the study design. Consequently, this limited the generalizability of the findings to broader contexts.

Another limitation was participant non-response. Some individuals declined to take part in the study, potentially introducing response bias. Non-respondents may differ systematically from those who participated, possibly leading to an underestimation of the actual prevalence of adequately iodized salt use. To mitigate this, adjustments for non-response were made during analysis. Additionally, hamlet leaders assisted in encouraging participation by clearly explaining the study's objectives and the potential benefits of involvement.

Data collection also encountered practical challenges. These included the frequent absence of household heads during the day, language barriers, and the risk of social desirability bias. To address these issues, several strategies were implemented. Households were revisited in the evenings when respondents were more likely to be at home. Hamlet leaders provided translation support and facilitated communication where language barriers existed.

To reduce social desirability bias, data collectors made efforts to build rapport with participants before asking sensitive questions. This helped create a trusting and comfortable environment, encouraging more honest responses. Participants were explicitly informed that their genuine answers were essential to the accuracy and success of the study. They were reassured that there were no right or wrong responses and that honesty was valued over socially acceptable answers.

Despite these limitations, the study offers valuable insights into the utilization of adequately iodized salt and the factors influencing its use, particularly in rural areas with local salt producers. These findings contribute to the existing body of knowledge and can inform future interventions aimed at improving iodine nutrition in similar settings.

## Conclusion

The utilization of adequately iodized salt in Kilwa District remains low, influenced by factors such as family size, knowledge gaps, improper storage, exposure to sunlight, and prolonged storage time. Local salt producers also contribute to poor availability due to non-compliance with iodization standards.

To address this, the Ministry of Health should implement targeted awareness campaigns on the benefits and proper handling of iodized salt. The CHMT and TFNC must also strengthen enforcement of regulations prohibiting the sale of non-iodized salt. Securing production sites in regions like Lindi is essential to ensure consistent supply and prevent theft.

At the community level, engaging village health workers can promote behavior change and increase household-level use of iodized salt.

A coordinated approach involving regulation, education, and supply chain protection is vital to eliminate iodine deficiency.

Furthermore, education plays a crucial role in changing household behaviors and increasing the acceptance and consistent use of iodized salt. Enhancing community based educational initiatives, including school programs and local health talks, can effectively improve awareness and dispel misconceptions about iodized salt. Incorporating education strategies into national policies can create a sustained impact on iodized salt utilization.

In addition to these strategies, developing a comprehensive national policy on iodized salt utilization is essential to guide government actions aimed at improving iodized salt use. Such policies should focus on strengthening regulatory frameworks, supporting small-scale producers, and ensuring equitable access across communities. Implementing clear, actionable policies will enhance the effectiveness of interventions and facilitate the long-term goal of eliminating iodine deficiency in the region. Future research should explore the salt value chain, challenges faced by small-scale producers, and household beliefs influencing usage to inform more tailored interventions.

## Supporting information

**S1 Data. Dataset used for the analysis of utilization of adequately iodized salt and its associated factors in rural areas of Kilwa District, Lindi Region, Tanzania, 2023. This file contains anonymized individual-level data used to generate all results presented in the manuscrip.**
(XLSX)

## Acknowledgments

We would want to thank everyone who took the time to ensure that this task was completed. I appreciate the heads of households from Kilwa district who participated in this study and the Ward Executive Officers from five wards for their assistance during data collection. I'm also grateful to Mr. Adolf Kalombo and Joseph Kimambo for assisting me in collecting the data at the field.

## Author contributions

**Conceptualization:** David Mahwera, Rose Msaki, Rogath Kishimba, Fatma Abdallah, Vicent Assay, Geofrey Mchau, Germana Leyna, Theresia Ambrose.

**Data curation:** David Mahwera, Geofrey Mchau.

**Formal analysis:** David Mahwera, Rose Msaki, Geofrey Mchau, Theresia Ambrose.

**Investigation:** David Mahwera.

**Methodology:** David Mahwera, Rogath Kishimba, Fatma Abdallah, Danford Mahwera, George Mrema, Geofrey Mchau, Theresia Ambrose.

**Supervision:** Rogath Kishimba, Geofrey Mchau, Germana Leyna, Theresia Ambrose.

**Validation:** David Mahwera, Rose Msaki, Rogath Kishimba, Fatma Abdallah, Danford Mahwera, Vicent Assay, George Mrema, Geofrey Mchau, Germana Leyna, Theresia Ambrose.

**Visualization:** David Mahwera, Rose Msaki, Rogath Kishimba, Fatma Abdallah, Danford Mahwera, Vicent Assay, George Mrema, Geofrey Mchau, Germana Leyna, Theresia Ambrose.

**Writing – original draft:** David Mahwera, Rose Msaki, Rogath Kishimba, Fatma Abdallah, Danford Mahwera, Vicent Assay, George Mrema, Geofrey Mchau, Germana Leyna, Theresia Ambrose.

**Writing – review & editing:** David Mahwera, Rogath Kishimba, Fatma Abdallah, Danford Mahwera, Vicent Assay, George Mrema, Geofrey Mchau, Germana Leyna, Theresia Ambrose.

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
