## [Decision Letter · Decision Letter 0]

10 Mar 2025

Dear Dr. Mrema,

Thank you for submitting your manuscript to PLOS ONE. After careful consideration, we feel that it has merit but does not fully meet PLOS ONE’s publication criteria as it currently stands. Therefore, we invite you to submit a revised version of the manuscript that addresses the points raised during the review process.

**ACADEMIC EDITOR:**

We look forward to receiving your revised manuscript.

Kind regards,

Charles Odilichukwu R. Okpala, PhD

Academic Editor

PLOS ONE

Journal Requirements:

2. Thank you for stating the following financial disclosure: This work is supported by the United States President’s Emergency Plan for AIDS Relief (PEPFAR) through the Centre of Excellence in Health Monitoring and Evaluation, Mzumbe University under the U.S Centers for Disease Control and Prevention (CDC), Project Cooperative Agreement No: NU2GGH002292. Additionally, they provide support in the training of residents in Tanzania's Field Epidemiology and Laboratory Training Program. 

Additional Editor Comments:

Major revisions required

Reviewers' comments:

Reviewer's Responses to Questions

**Comments to the Author**

1. Is the manuscript technically sound, and do the data support the conclusions?

Reviewer #1: Yes

Reviewer #2: Partly

2. Has the statistical analysis been performed appropriately and rigorously?

Reviewer #1: Yes

Reviewer #2: Yes

3. Have the authors made all data underlying the findings in their manuscript fully available?

Reviewer #1: Yes

Reviewer #2: Yes

4. Is the manuscript presented in an intelligible fashion and written in standard English?

Reviewer #1: Yes

Reviewer #2: Yes

Reviewer #1: he manuscript presents important findings with public health implications. However, some revision need to improve it.

Language and Typographical Errors: Correct grammatical issues such as (e.g., "expose of salt" → "exposure of salt"; "AnIodine rapid test" → "An iodine rapid test").

• The introduction ends with a detailed explanation of the study's purpose, but it could be more concise. For example, the sentence "By knowing the current prevalence of households utilizing iodized salts, factors associated with iodized salt utilization and reasons for adequate availability of iodized salt in the community among local salt producers in Kilwa, the government through TFNC will be able to design appropriate intervention to improve the coverage of household and to help local salt producers to produce salt which is adequately iodized" is quite long and could be broken into smaller, more digestible sentences. In addition, the introduction provides good background information, but it should more clearly outline the research gap

Methodology:

Provide more detail on the iodine testing methodology

Disscusion

The discussion jumps between different findings and comparisons without a clear transition. For instance, the section on family size (lines 301–311) is followed by a discussion on knowledge (lines 312–318), then storage practices (lines 319–328), and then time to stay with salt (lines 329–334). While these are all relevant, the flow could be improved by grouping related findings together

Reviewer #2: Comments and recommendations for the author:

Abstract:

(a) Your abstract should adhere to the journal's format, presented as a single continuous paragraph while incorporating all necessary components

Introduction:

(a) Please ensure consistency in the WHO iodine intake recommendation unit—whether it is in micrograms or grams. Kindly review and revise the reference statement accordingly in lines 51-54 and throughout the manuscript.

Study Design & Justification:

(a) Why was a cross-sectional design chosen for this study, and how does it adequately address the study objectives

(b) Were any potential biases considered in selecting the Kilwa district as the study area

Sampling Methodology:

(a) How was the sample size of 493 households determined? Were design effects or response rates considered

(b) Can the authors provide more justification for the choice of systematic sampling and how it ensures representativeness

(c) What steps were taken to minimize selection bias, particularly for the qualitative component

Data Collection & Quality Control:

(a) Were there any challenges encountered during data collection, and how were they mitigated

(b) How was the accuracy of iodine measurement ensured, particularly for the WYD machine analysis

(c) Can the authors clarify whether the same research assistants collected both qualitative and quantitative data? If so, were they trained to avoid interviewer bias

Data Analysis:

(a) What measures were taken to address missing data, if any

(b) Why was a p-value threshold of 0.2 used for bivariable analysis inclusion

(c) Were any adjustments made for clustering in the two-stage sampling approach

Ethical Considerations:

(a) How was confidentiality maintained during in-depth interviews, considering the sensitivity of some responses

(b) Were participants informed about their right to withdraw from the study at any time without consequences

Clarity and Rationale

(a) The study highlights a significantly low proportion (9.4%) of households utilizing adequately iodized salt. Could the authors provide more context on national or regional policies regarding salt iodization and enforcement?

(b) The discussion compares findings to various studies in Ethiopia and India. Why were these specific regions chosen for comparison? Are there studies within Tanzania or East Africa that might offer more relevant comparisons

Methodology

(a) What was the sampling technique used to select the 435 households? Was it random sampling, stratified sampling, or another method

(b) The study mentions binary and multivariable logistic regression analyses. Could the authors clarify if they conducted any model diagnostics (e.g., goodness-of-fit tests) to assess the reliability of their findings

(c) The manuscript states that cultural beliefs were associated with salt utilization in the binary regression but were not significant in the multivariable model. Could the authors elaborate on potential confounders that might have led to this change in significance

Findings and Interpretation

(a) The study identifies a strong association between salt storage conditions and iodine retention. Were other household practices, such as cooking methods (e.g., adding salt before or after boiling), considered as potential factors

(b) The role of knowledge about iodized salt is emphasized in the discussion. Did the study assess the sources of knowledge (e.g., health campaigns, media, school education) to better understand how information dissemination could be improved

Limitations and Recommendations

(a) Were there any limitations regarding data collection, such as social desirability bias, especially in self-reported knowledge about iodized salt

(b) The authors recommend working with local salt producers to ensure a reliable iodized salt supply. Have there been any past interventions in Tanzania to improve iodized salt availability, and how effective were they

Recommendations for Improvement:

1. Clarify Study Area Selection:

Provide more details on why Kilwa was considered the "worst-performing" district in Lindi beyond citing an unpublished TFNC report. If possible, reference publicly available data.

2. Improve Sampling Justification:

Explain why different selection intervals were used for different wards.

Justify the choice of convenience sampling for local production sites and discuss potential limitations.

3. Enhance Methodological Transparency:

Provide more information on how qualitative data saturation was determined.

Clarify whether data collectors were blinded to the iodine content of the salt samples to reduce potential bias.

4. Strengthen Data Analysis Explanation:

Explain whether and how potential confounders were accounted for in the logistic regression model.

Provide a rationale for why NVivo was chosen for qualitative data analysis over other possible methods.

5. Address Ethical and Practical Considerations:

Explicitly state whether informed consent was obtained from all participants for audio recording.

Discuss how potential risks to participants were minimized, especially for local salt producers.

6. Data Presentation & Clarity

The manuscript refers to Figure 2 but does not describe it in detail. A more explicit explanation of the figure’s content would improve readability.

7. Methodology Enhancement

If feasible, provide details on how "good knowledge" about iodized salt was assessed. Was there a standardized questionnaire or knowledge scale used?

8. Further Discussion and Implications

Given the cultural preference for non-iodized salt, it would be useful to suggest culturally sensitive education strategies that address local misconceptions.

The study highlights financial barriers to iodized salt use. Can the authors discuss possible government subsidies or pricing strategies to improve affordability?

9. Policy and Public Health Recommendations

Advocate for stricter enforcement of salt iodization policies, similar to Nepal's Iodized Salt Act, as a potential regulatory approach for Tanzania.

Suggest community-level initiatives, such as engagement with village health workers, to educate and promote adequately iodized salt utilization.

**Do you want your identity to be public for this peer review?** For information about this choice, including consent withdrawal, please see our Privacy Policy

Reviewer #1: **Yes: ** NUHA Al-Aghbari

Reviewer #2: No

---

## [Author Response · Author response to Decision Letter 1]

25 Jun 2025

We express our gratitude to the editor and reviewers for their valuable comments, which have significantly enhanced our manuscript titled " Utilization of adequately iodized salt and its associated factors in Tanzania rural areas: a case of Kilwa district, Lindi region, 2023". We have considered all the comments and integrated their suggestions into the revised version of the manuscript. We believe that we have effectively addressed the raised concerns and that, after incorporating the changes, our manuscript is now appropriate for publication.

---

## [Decision Letter · Decision Letter 1]

14 Jul 2025

Dear Dr. Mrema,

Thank you for submitting your manuscript to PLOS ONE. After careful consideration, we feel that it has merit but does not fully meet PLOS ONE’s publication criteria as it currently stands. Therefore, we invite you to submit a revised version of the manuscript that addresses the points raised during the review process.

**ACADEMIC EDITOR:  ***(Carefully adhere to instructions given below)*****

We look forward to receiving your revised manuscript.

Kind regards,

Charles Odilichukwu R. Okpala, PhD

Academic Editor

PLOS ONE

Journal Requirements:

Additional Editor Comments (if provided):

Authors, thank you for your patience. Reviewers have considered your revised manuscript for publication. Please, before it can be accepted, some additional touches are needed, as below:

a) In the introduction, a number of places need to improvement. Firstly, the objective must be clearly stated as the very last sentence " To supplement existing literature, therefore, this current work assessed household utilization of adequately iodized salt and its associated factors in Kilwa district, which is among Tanzania rural areas, and this was performed by way of cross-sectional design across local salt producers" (Please insert this sentence in current line 86).

Now, let's look at the overall introduction. It is not yet complete. Merge line 48-63 as one paragraph. Also, merge lines 77-86 as one paragraph. This will give 4 paragraphs in total. Before talking about USI in Tanzania, please, create a new paragraph to talk about salt production process in Tanzania. Tell us the steps of salt production, tell us the number of salt makers in Tanzania, tell us how their packaging looks like, tell us how many people generally work in a typical local salt producer, tell us their pattern of salt product distribution, tell us about affordability of salt (sources of this information could be newspapers, government bulletins, personal communication), I expect this will likely be one strong paragraph, at the maximum two strong paragraphs

b) Go to the methods,change the title of "2. Study methodology" (not "Materials and methods"). Now, begin this section by creating a new subsection called "Schematic overview of study program", which would comprise 5 sentences, and supported compulsorily by a flow diagram that shows the major methodology steps that built the entire study. Sentence one must introduce the flow diagram, sentence two briefly outlines the major stages, sentence 3 links the flow diagram with the objective of this work, sentence 4 tells us why this approach is unique, sentence 5 tells us how this steps were validated.

After this section called "Schematic overview of study program", the next section must be "Questionnaire development process". Tell us, how did you develop the questionnaire (the research instrument)? how did you use specialist experience/expertise and knowledge together with synthesised relevant literature to come up with the questionnaire items? How did you develop the questionnaire contents, and did you do content validation? After the validation process, who subsequently approved the questionnaire? Is it the association body for the local salt producers (just curious)? Was there any prerequisite conditions set out for questionnaire participation? Tell us about the arrangement of the actual questionnaire, how many pages, how many items in total, and the major sections. (Make sure to include aspects of this in the flow diagram, and mention it tinyly in the " "Schematic overview of study program" section) . How did you differentiate between Semi Structured and pre-tested questionnaires? what was the main purpose to do so?

c) Please amend "Study design, area and population under the study" to "Study design, area and population"

The rest of the 2. Study methodology" is ok

d) Results and discussion are excellent. Please, in the discussion, make best effort to capture all the tables mentioned in the results. Use "(Refer to Table ?) in all the places where data is each table is discussed

e) Before conclusion, I believe that there were limitations that you encountered, please create a new section, "Challenges and limitations of current study", make sure to brainstorm on limitations you encountered in developing the questionnaire, developing the entire methods, assembling the data, analyzing the data, discussing the findings, logistical challenges, administrative challenges, etc. Maximum, two paragraphs. Authors apply your kind discretion to develop this section.

f) in the conclusion, provide clear direction for future study. Also, what is the take home lesson for policy makers and other stakeholders.

Look forward to your revised manuscript. Please, attend to every single bit of these comments. The editor will thoroughly examine every single point one by one. I believe if all are diligently implemented, this work will be of higher quality.

Reviewers' comments:

Reviewer's Responses to Questions

**Comments to the Author**

Reviewer #1: All comments have been addressed

Reviewer #2: All comments have been addressed

2. Is the manuscript technically sound, and do the data support the conclusions?

Reviewer #1: Yes

Reviewer #2: Yes

3. Has the statistical analysis been performed appropriately and rigorously?

Reviewer #1: Yes

Reviewer #2: Yes

4. Have the authors made all data underlying the findings in their manuscript fully available?

Reviewer #1: Yes

Reviewer #2: Yes

5. Is the manuscript presented in an intelligible fashion and written in standard English?

Reviewer #1: Yes

Reviewer #2: Yes

Reviewer #1: Thank you for addressing my comments thoroughly. I appreciate that your response aligns with the suggestion I provided. I’m glad to see the revisions have been made accordingly, and I have no further concerns regarding this point.

Wishing you all the best in the publication process.

Reviewer #2: The authors have significantly improved their manuscript on iodized salt utilization in Tanzania's Kilwa district through comprehensive revisions. They clarified methodological details, strengthened discussions with refined comparisons and policy implications, improved result presentations, addressed ethical considerations and biases, and provided actionable recommendations including culturally sensitive education strategies. These revisions resolve all major concerns and meet journal standards. I recommend acceptance for publication in PLOS ONE.

**Do you want your identity to be public for this peer review?** For information about this choice, including consent withdrawal, please see our Privacy Policy

Reviewer #1: **Yes: ** NUHA AMER Al-Aghbari

Reviewer #2: **Yes: ** Jonah Bawa Adokwe PhD

---

## [Author Response · Author response to Decision Letter 2]

26 Aug 2025

We express our gratitude to the editor for their valuable comments, which have significantly improved our manuscript titled " Utilization of adequately iodized salt and its associated factors in Tanzania rural areas: a case of Kilwa district, Lindi region, 2023". We have carefully revised our manuscript addressing all comments and suggestions provided by the editor. We believe that we have effectively addressed the raised concerns and that, after incorporating the changes, our manuscript is now appropriate for publication.

---

## [Decision Letter · Decision Letter 2]

24 Sep 2025

Dear Dr. Mrema,

Thank you for submitting your manuscript to PLOS ONE. After careful consideration, we feel that it has merit but does not fully meet PLOS ONE’s publication criteria as it currently stands. Therefore, we invite you to submit a revised version of the manuscript that addresses the points raised during the review process.

**ACADEMIC EDITOR:**

We look forward to receiving your revised manuscript.

Kind regards,

Charles Odilichukwu R. Okpala, PhD

Academic Editor

PLOS ONE

Journal Requirements:

Additional Editor Comments:

Please, kindly attend to the comments raised as diligently and detailed as possible

Reviewers' comments:

Reviewer's Responses to Questions

**Comments to the Author**

Reviewer #3: (No Response)

2. Is the manuscript technically sound, and do the data support the conclusions?

Reviewer #3: Partly

3. Has the statistical analysis been performed appropriately and rigorously?

Reviewer #3: Yes

4. Have the authors made all data underlying the findings in their manuscript fully available?

Reviewer #3: Yes

5. Is the manuscript presented in an intelligible fashion and written in standard English?

Reviewer #3: Yes

Reviewer #3: Utilization of adequately iodized salt and its associated factors in Tanzania rural areas:

a case of Kilwa district, Lindi region, 2023

This is quite a good research report. Although the study was conducted on a small scale, it is meaningful and necessary for many regions in African countries where people’s awareness of the importance of iodized salt in food remains low. The study also collected a relatively large sample size, sufficient for analysis and assessment of iodized salt utilization in the research area. The information presented in the supplementary materials is also quite interesting and diverse. The methodology and data analysis methods applied are appropriate.

However, the research results have not been evaluated and discussed from multiple perspectives, which shows that the study has not fully exploited the data collected. In particular, the information collected from 4 wards (namely: Kiranjeranje, Mandawa, Miteja, and Tingi), further divided into 12 villages (namely: Kiranjeranje, Kiswele, Mbwemkuru, Mtandi, Mandawa, Mitumba, Hotel tatu, Kiwawa, Sinza, Matandu, Kibaoni, Njia nne, and Mtandango), has not been analyzed in detail at the smaller administrative unit level. The study results are presented only in Figure 1, which is too simple and sketchy. The paper should include more results in the form of charts to demonstrate thorough data processing, especially results related to educational level, gender, age, and iodized salt usage.

I have some concerns regarding the proportion of survey respondents. The age group 26–35 accounted for a very large number of respondents (252), while other age groups were underrepresented. In particular, the group under 25 years old was very limited (only 16 respondents under 18 and 28 respondents aged 18–25). This group is crucial for nutritional development and represents the future of the country, so such a small number of responses may lead to results that are not truly representative. Could you clarify whether, among the survey participants, there were any individuals suffering from iodine deficiency-related diseases?

The conclusion section should include additional policy recommendations to help the government address this issue. I believe that education is also an important aspect, which was mentioned at the beginning of the paper but not discussed in the conclusion. If possible, a separate section on policy recommendations for the government to improve iodized salt utilization should be developed, which would make the paper much more practically valuable.

**Do you want your identity to be public for this peer review?** For information about this choice, including consent withdrawal, please see our Privacy Policy

Reviewer #3: **Yes: ** Trung Quang Nguyen

---

## [Author Response · Author response to Decision Letter 3]

5 Nov 2025

We have considered all the comments and integrated their suggestions into the revised version of the manuscript. We believe that we have effectively addressed the raised concerns and that, after incorporating the changes, our manuscript is now appropriate for publication.

---

## [Editor Report · Decision Letter 3]

6 Nov 2025

Utilization of adequately iodized salt and its associated factors in Tanzania rural areas: a case of Kilwa district, Lindi region, 2023

PONE-D-25-02467R3

Dear Dr. Mrema,

We’re pleased to inform you that your manuscript has been judged scientifically suitable for publication and will be formally accepted for publication once it meets all outstanding technical requirements.

Kind regards,

Charles Odilichukwu R. Okpala, PhD

Academic Editor

PLOS ONE

Additional Editor Comments (optional):

Thank you very much for your revised work. Accepted for publication
---

## [Editor Report · Acceptance letter]

PONE-D-25-02467R3

PLOS ONE

Dear Dr. Mrema,

I'm pleased to inform you that your manuscript has been deemed suitable for publication in PLOS ONE. Congratulations! Your manuscript is now being handed over to our production team.

Kind regards,

on behalf of

Dr. Charles Odilichukwu R. Okpala

Academic Editor

PLOS ONE